# Using strain to uncover the interplay between two- and three-dimensional charge density waves in high-temperature superconducting YBa₂Cu₃Oy

I. Vinograd[1,2,10], S. M. Souliou[1,10], A.-A. Haghighirad[1], T. Lacmann[1], Y. Caplan[3], M. Frachet[1], M. Merz[1,4], G. Garbarino[5], Y. Liu[6], S. Nakata[6], K. Ishida[7,9], H. M. L. Noad[7], M. Minola[6], B. Keimer[6], D. Orgad[3], C. W. Hicks[7,8] & M. Le Tacon[1] ✉

Uniaxial pressure provides an efficient approach to control charge density waves in YBa₂Cu₃Oy. It can enhance the correlation volume of ubiquitous short-range two-dimensional charge-density-wave correlations, and induces a long-range three-dimensional charge density wave, otherwise only accessible at large magnetic fields. Here, we use x-ray diffraction to study the strain dependence of these charge density waves and uncover direct evidence for a form of competition between them. We show that this interplay is qualitatively described by including strain effects in a nonlinear sigma model of competing superconducting and charge-density-wave orders. Our analysis suggests that strain stabilizes the 3D charge density wave in the regions between disorder-pinned domains of 2D charge density waves, and that the two orders compete at the boundaries of these domains. No signatures of discommensurations nor of pair density waves are observed. From a broader perspective, our results underscore the potential of strain tuning as a powerful tool for probing competing orders in quantum materials.

A tendency towards charge ordering in the underdoped high-temperature superconducting cuprates was predicted soon after their discovery[1–3]. However, it took decades of effort with a variety of experimental methods to demonstrate this phenomenon and its ubiquity across the cuprate superconductors[4–12]. This effort has raised many new questions. One observation that lacks a clear explanation is the fact that the competition between charge order and superconductivity is striking in some cuprate families[9,10,13], but is less pronounced in others[11,12,14]. Another is the very substantial variation in correlation length, which varies over two orders of magnitude amongst the cuprates families − from a few unit cells to several tens of nm − without any obvious relation to the superconducting critical temperature $T_c$. To address these issues, it has proved useful to focus on materials with low disorder scattering, and to probe them with

[1]Institute for Quantum Materials and Technologies, Karlsruhe Institute of Technology, Kaiserstr. 12, D-76131 Karlsruhe, Germany. [2]4th Physical Institute - Solids and Nanostructures, University of Göttingen, D-37077 Göttingen, Germany. [3]Racah Institute of Physics, The Hebrew University, Jerusalem 91904, Israel. [4]Karlsruhe Nano Micro Facility (KNMFi), Karlsruhe Institute of Technology, Kaiserstr. 12, D-76131 Karlsruhe, Germany. [5]ESRF, The European Synchrotron, 71, avenue des Martyrs, CS 40220, F-38043 Grenoble Cedex 9, France. [6]Max Planck Institute for Solid State Research, Heisenbergstraße 1, D-70569 Stuttgart, Germany. [7]Max Planck Institute for Chemical Physics of Solids, Nöthnitzer Str. 40, D-01187 Dresden, Germany. [8]School of Physics and Astronomy, University of Birmingham, Birmingham B15 2TT, UK. [9]Present address: Institute for Materials Research, Tohoku University, Sendai 980-8577, Japan. [10]These authors contributed equally: I. Vinograd, S. M. Souliou. ✉e-mail: matthieu.letacon@kit.edu

external tuning parameters, such as magnetic field[10] or pressure[15–17], that do not introduce disorder.

While hydrostatic pressure has been widely used to investigate unconventional superconductors and correlated electron systems, recent studies have highlighted several benefits of uniaxial stress[18–21]. In the cuprates, uniaxial stress has been used to probe charge stripes in La-based compounds[22–24], and to polarize them[25,26]. Interestingly and in contrast to the introduction of pair-breaking impurities[27] or vortices under high magnetic fields[10], uniaxial stress can also be used to homogeneously suppress $T_c$[28]. This allows alternative electronic orders to develop, and the interplay between these orders and the super-conductivity to be studied with precision.

Uniaxial compression along the $a$ axis (the shortest axis of the orthorhombic structure) has been shown to suppress $T_c$ by up to 30%[28] in $YBa_2Cu_3O_{6.67}$. In this compound, this results in an enhancement of the amplitude and correlation lengths of an incommensurate, short-range, 2D-correlated charge density wave (hereafter referred to as the 2D CDW)[29,30].

Beyond a threshold stress value, a long-range-ordered, 3D-correlated CDW (hereafter referred to as the 3D CDW) emerges[29,30]. 3D CDW order can also be induced by a magnetic field, though to date, with lower amplitude and shorter correlation lengths than the one induced by uniaxial stress[8,31–35]. The 2D CDW is biaxial, with components that propagate along both the $a$ and $b$ axes, while the 3D CDW is uniaxial, propagating along the $b$ axis only. On the other hand, the in-plane wavelengths of the 2D and 3D CDWs are identical[29,32] and the analysis of the in-plane correlation lengths revealed that the 2D CDW consists of quasi-independent unidirectional orders[36]. This has further been confirmed by the observation of substantially different dependencies of amplitude and correlation lengths of its $a$ and $b$ components on uniaxial stress[30]. Overall, this suggests that the 2D and 3D CDWs might be intimately related.

A key question remains to be clarified: does the 3D CDW emerge from "patches" of 2D CDWs that lock together, or is it a separate order, formed by different charge carriers, whose periodicity locks to that of the background 2D CDWs? Additionally, there is so far little information on where the 3D CDW exists in strain-temperature space, which is crucial information for understanding its interaction with the superconductivity.

Here, we tackle this issue in underdoped $YBa_2Cu_3O_y$ with synchrotron hard x-ray diffraction under uniaxial stress. We show that the 2D CDW amplitude stops growing when the 3D CDW onsets, and map the boundaries of the 3D CDW phase in a strain-temperature phase diagram. We further report that the strain-induced formation of the 3D CDW and its interplay with the 2D CDW and superconductivity are qualitatively well described by including the effects of strain in an extension of a nonlinear sigma model[37] previously used to investigate the magnetic-field-induced formation of the 3D CDW order in YBCO[38,39]. In this picture, the long-range 3D CDW emerges from the regions between disorder-induced 2D CDW domains (*i.e.*, the regions where superconductivity forms in the absence of strain). The growth of disordered 2D CDW domains is then halted by their increased surface tension due to phase-mismatch at the boundary with phase-ordered 3D CDW. Finally, we have also looked for signs of discommensurations, which would indicate a tendency of the CDW to lock to the lattice[40], and for pair–density wave (PDW) correlations[41] on top of the 3D CDW, but have found no evidence for either.

## Results

Hard x-ray diffraction was performed as a function of temperature on uniaxially pressurized. $YBa_2Cu_3O_y$ ($YBCO_y$) single crystals, as described in the Methods section. We first discuss the effect of uniaxial compression along the $a$ and $b$ directions on the 2D CDW, for samples with $y = 6.67$ (corresponding to a doping level $p ≈ 0.125$). As shown in the intensity maps in Fig. 1, the $b$ component of the 2D CDW

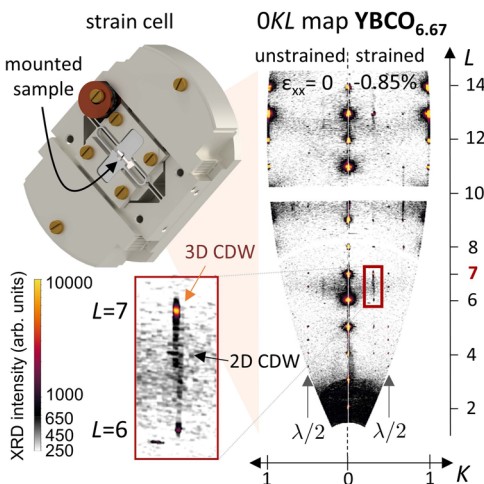

**Fig. 1 | Experimental setup and representative results of x-ray scattering under uniaxial stress.** Measurements were performed in transmission geometry using a Razorbill CS200T strain cell. The $YBCO_{6.67}$ sample was mounted onto an exchangeable titanium cross. Sections of large $(0KL)$ reciprocal space maps at zero (left) and highest $a$ axis compression (right) show the appearance of sharp 3D CDW peaks at integer $L$. Faint 2D CDW intensity in between is present at all strains at half-integer $L$ on top of a diffuse phonon background. Grey arrows mark sharp Bragg reflections that appear at $k = 0.5$ due to higher harmonics with $λ/2$ (see Methods section for the wavelength $λ$). The inset shows an enlarged part of the $(0KL)$ map. To make the 2D CDW more apparent, the colour scale is linear for small and loga-rithmic for large intensities.

has an extremely broad profile along the $[0, 0, L]$ direction, centered at half-integer $L$, but is relatively sharp along the $[0, K, 0]$ direction already in the absence of strain. The incommensurate wavevector of the modulations are in perfect agreement with previous reports[9,10,27,42]. To evaluate the intensity along the $K$ direction, we average the intensity over the range $0.4 < L < 0.6$, with results at a few $a$ axis strain values ($ε_{xx}$) shown in Fig. 2a. For these measurements, the temperature is close to $T_c(ε_{xx})$. For strains not exceeding the 3D CDW onset strain, the 2D CDW is most intense at $T ≈ T_c(ε_{xx})$. It can be seen in Fig. 2a that the intensity of the $b$ component of the 2D CDW approximately doubles between $ε_{xx} = 0$ and $-0.55\%$ (but the CDW ordering wave vector is only weakly affected by strain). The inte-grated intensity at a denser set of strains is shown in Fig. 2b, in which this doubling is again visible, along with a reduction in intensity for $ε_{xx} < -0.55\%$. Also shown in Fig. 2b is the integrated intensity of the $b$ component under compression along the $b$ axis ($ε_{yy} < 0$). The effect of $b$ axis compression is opposite to that of $a$ axis compression: the intensity shrinks. In Supplementary Fig. 5 we provide evidence con-firming the growth of the $a$ axis component of the 2D CDW under $b$ axis compression, in agreement with previous findings[30]. Together, these data show that the $a$- and $b$-CDWs are quasi-independent order parameters, and emphasize the uniaxial nature of the underlying order parameter[30,36].

The decrease in intensity of the 2D CDW for $ε_{xx} < -0.55\%$ coin-cides with the onset of the 3D CDW, which manifests itself as a sharp peak at integer $L$, signaling long-range, in-phase correlations along the $c$ axis[32,33]. The 3D CDW intensity is also shown in Fig. 2b. It is worth noting that this onset strain is considerably lower than that estimated in previous x-ray scattering studies[29,30], in which the lattice parameter along the direction of compression was not accessible and in which strain could therefore not be determined with sufficient accuracy. It is however closer to the location of an anomaly identified in the stress dependence of $T_c$ in ref. 28. The 3D CDW peak is also seen for $y = 6.55$ ($p = 0.108$) (see below), but not yet for $y = 6.80$ ($p = 0.140$), where, admittedly, the largest strain achieved in the present work was lower: we reached $ε_{xx} = -0.45\%$ (see Supplementary Fig. 6).

We turn next to the temperature dependence of the 2D and 3D CDW intensities. Integrated intensities of the 2D and 3D CDWs versus temperature at selected, fixed strains are shown in Fig. 3a for YBCO$_{6.67}$, and Fig. 3b for YBCO$_{6.55}$. At both of the selected strains for both

compositions, the 2D CDW intensity stops increasing (and within experimental accuracy even seems to decrease) when the 3D CDW onsets. At these selected strains, the onset temperature of the 3D CDW exceeds $T_c(\varepsilon_{xx} = 0)$ and so, presuming that $a$ axis compression suppresses $T_c$ in YBCO$_{6.55}$ as it does in YBCO$_{6.67}$ (and as inferred from thermal expansion in the zero stress limit[43]), it substantially exceeds $T_c$ at these strains. Therefore, the end of the 2D CDW growth at these strains is clearly tied to the onset of the 3D CDW, and not to the onset of superconductivity as at $\varepsilon_{xx} \sim 0$.

We summarize our findings in the form of intensity maps of the 3D CDW in strain-temperature space, shown for YBCO$_{6.67}$ and YBCO$_{6.55}$ in Fig. 3c, d, respectively. For YBCO$_{6.55}$ there is no direct measurement of $T_c(\varepsilon_{xx})$, so within the 3D CDW phase we estimate this quantity from the temperature where the 3D CDW signal intensity reaches a maximum, before getting gradually suppressed. This indicates a substantial range of overlap between the 3D CDW and superconductivity. It is, on the one hand, consistent with measurements in high magnetic fields that systematically find the 3D CDW onset field to lie below $H_{c2}$[33], and on the other hand with the 3D CDW signal being observed down to at least 41 K, i.e., well below $T_c(\varepsilon_{xx})$ for YBCO$_{6.67}$ at the highest measured strain of ref. 29 (the present data allow us to re-estimate this value to be $\varepsilon_{xx} = -0.85\%$). For YBCO$_{6.55}$, the 3D CDW onset lies at a slightly smaller strain, $\varepsilon_{xx} = -0.4\%$, lower than that for YBCO$_{6.67}$.

We end this section with a description of structural changes induced by the 3D CDW formation which can be quantified, despite experimental limitations on the accessible part of the reciprocal space imposed by the strain cell. Within experimental uncertainty, we observe the main structural changes in the CuO$_2$-planes as the samples are strained. From structural refinements of YBCO$_{6.67}$ we observe that the clearest response of the average structure (single unit cell) to the 3D CDW onset with increasing strain is a growing anisotropy of the buckling angle of the planar oxygen bonds as shown in Supplementary Fig. 7. Minor changes can be discerned in the parameters of the planar Cu atoms, e.g., in the intra-bilayer distance, but these are less evident than the change observed for the buckling angles of the oxygen bonds. Nevertheless, smaller motions are to be expected for the heavier Cu atoms. This also finds confirmation in the smaller isotropic temperature factors, shown in Supplementary Fig. 8.

## Model
In an effort to elucidate the experimental findings we consider a nonlinear sigma model describing competition between

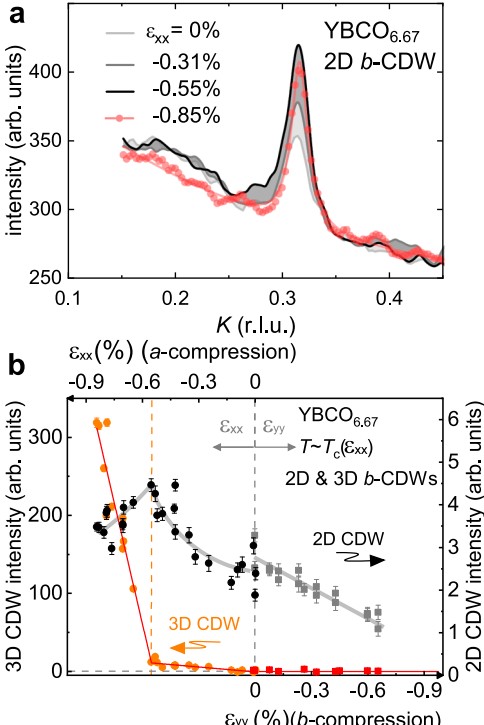

**Fig. 2 | Strain dependence of 2D and 3D CDW intensity for YBCO$_{6.67}$.** **a** 2D $b$-CDW intensity from $K$-cuts at half-integer $L$ values for $a$ axis compression. $a$ axis compression amplifies the $b$-CDW only up to the onset of 3D ordering. **b** $a$ axis and $b$ axis compression dependence of the integrated 2D CDW and 3D CDW intensities from $K$-cuts at half-integer $L = 6.5$ through the ordering wavevector of the 2D CDW **q$_{2D\text{-}CDW}$** (black and grey symbols) and $L$-cuts through the ordering wavevector of the 3D CDW **q$_{3D\text{-}CDW}$** (orange and red symbols). Error bars correspond to standard deviations of Lorentzian (Gaussian) fits. The vertical scales of the integrated intensities (line cuts) at $T$ - $T_c(\varepsilon_{xx})$ correspond to the same total integrated intensity, $I_{tot}$, based on Supplementary Table 2. The horizontal grey dashed line lies at zero. The orange dashed line marks the onset of long-range 3D CDWs at $\varepsilon_{xx}$=-0.55%.

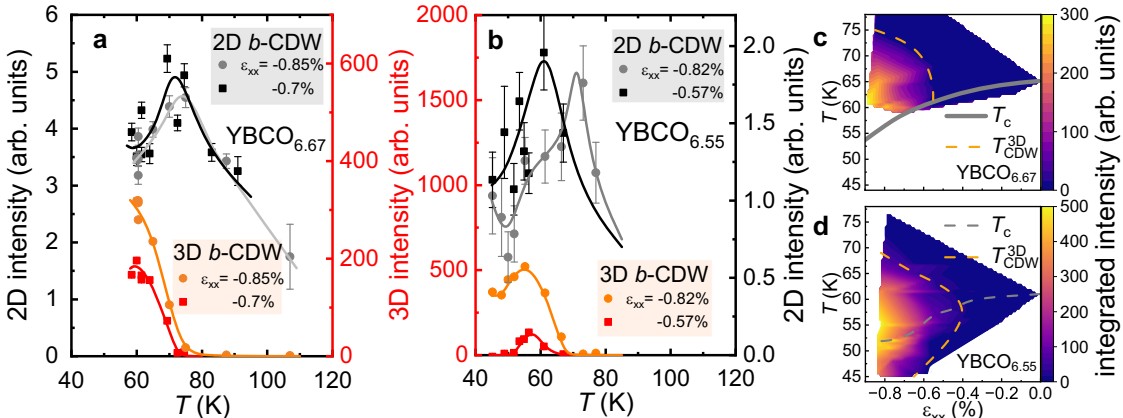

**Fig. 3 | Temperature dependence of 2D and 3D CDW intensity for YBCO$_{6.67}$ and YBCO$_{6.55}$.** **a** Temperature dependence of 3D and 2D CDW intensities for YBCO$_{6.67}$ at $a$ axis compressions $\varepsilon_{xx} = -0.7$ % and $-0.85$ %. **b** 3D and 2D CDW intensities for YBCO$_{6.55}$ at compressions of $\varepsilon_{xx} = -0.57$ % and $-0.82$ %. 2D CDW (3D CDW) intensities are determined from $K$-cuts ($L$-cuts). Error bars correspond to standard deviations of Lorentzian fits. Lines are guides to the eye. For a comparison of 2D and

3D intensities, see Supplementary Table 2. **c** Strain-temperature phase diagram of the 3D CDW intensity for YBCO$_{6.67}$ with the $T_c$ curve reproduced from ref. 28. Dashed orange lines mark the boundary of the 3D CDW phase. In the corresponding diagram for YBCO$_{6.55}$ in (**d**) $T_c(\varepsilon_{xx})$ is estimated from the linear extrapolation down to the onset of the 3D phase[43]. Within the 3D CDW phase $T_c(\varepsilon_{xx})$ is based on the peak in the 3D CDW intensity.

superconductivity and CDW orders[37–39]. It is formulated in terms of a three-dimensional complex vector whose components correspond to a superconducting order parameter, $\psi_{j\mu}(\mathbf{r})$, and two complex CDW order parameters, $\Phi_{j\mu}^{a,b}(\mathbf{r})$. The latter describe density variations $\delta\rho_{j\mu}(\mathbf{r}) = e^{i\mathbf{Q}_a\cdot\mathbf{r}}\Phi_{j\mu}^a(\mathbf{r}) + e^{i\mathbf{Q}_b\cdot\mathbf{r}}\Phi_{j\mu}^b(\mathbf{r}) + \text{c.c.}$, along the $a$ and $b$ directions with incommensurate wave-vectors $\mathbf{Q}_{a,b}$. The order parameters reside on bilayers that are indexed by $j$ and represent the CuO$_2$ bilayers of YBCO. The index $\mu = 0, 1$ denotes the bottom (top) layer and $\mathbf{r}$ is the in-plane position vector. In the following we will coarse grain each of the planes into a square lattice whose lattice constant is the observed CDW wavelength, i.e., about 3 Cu-Cu spacings. The model assumes the existence of some type of local order at every lattice point and incorporates the competition between the different components via the constraints

$$|\psi_{j\mu}|^2 + |\boldsymbol{\Phi}_{j\mu}|^2 = 1, \tag{1}$$

where $\boldsymbol{\Phi}_{j\mu} = (\Phi_{j\mu}^a, \Phi_{j\mu}^b)^T$.

The vector of order parameters is governed by the Hamiltonian

$$
\begin{aligned}
H = \sum_j \sum_{\mu=0,1} H_{j\mu} + \frac{\rho_s}{2} \sum_j \sum_{\mathbf{r}} \Big[ &\tilde{U}\boldsymbol{\Phi}_{j,0}^\dagger \boldsymbol{\Phi}_{j,1} \\
&+ U\boldsymbol{\Phi}_{j,1}^\dagger \boldsymbol{\Phi}_{j+1,0} - \tilde{J}\psi_{j,0}^* \psi_{j,1} - J\psi_{j,1}^* \psi_{j+1,0} \\
&+ V_j^\dagger \big(\gamma\boldsymbol{\Phi}_{j,0} + \boldsymbol{\Phi}_{j,1} + \boldsymbol{\Phi}_{j+1,0} + \gamma\boldsymbol{\Phi}_{j+1,1}\big) + \text{c.c.}\Big].
\end{aligned}
\tag{2}
$$

Henceforth, the bare superconducting stiffness, $\rho_s$, is set to 1 and serves as the basic energy scale. We model the Coulomb interaction between CDW fields within a bilayer by a local coupling $\tilde{U}$, and denote the intra-bilayer Josephson tunneling amplitude by $\tilde{J}$. The (weaker) Coulomb interaction and Josephson coupling between nearest-neighbor planes belonging to consecutive bilayers are denoted by $U$ and $J$, respectively. In the last term, we include the coupling between the disordered doped oxygens on the chain layers and the CDW fields on the adjacent bilayers. This coupling may originate from local changes in the concentration of doped holes and from the Coulomb interaction between the oxygens and the CDW[14]. Hence, we assume that it is reduced by a factor $\gamma$ as one moves from the inner to the outer CuO$_2$ planes. We model the effective potential induced by the chain layers as a collection of randomly placed discs at $\mathbf{r}_l$, each of radius $r_d$ and containing a constant potential with a random phase that couples either to the $a$ or $b$ CDW fields, $\mathbf{V}(\mathbf{r}) = V\sum_l f(|\mathbf{r} - \mathbf{r}_l|)e^{i\theta_l}(p_l, 1 - p_l)^T$. Here, $p$ takes the values 0,1 with probability 1/2 and $f(\mathbf{r}) = 1 - \Theta(|\mathbf{r}| - r_d)$, where $\Theta(r)$ is the step function.

Within a layer the Hamiltonian reads

$$
\begin{aligned}
H_{j\mu} = \frac{\rho_s}{2} \sum_{\mathbf{r}} \Big[ &|\boldsymbol{\nabla}\psi_{j\mu}|^2 + \lambda|\boldsymbol{\nabla}\boldsymbol{\Phi}_{j\mu}|^2 + g|\boldsymbol{\Phi}_{j\mu}|^2 \\
&+ (\Delta g - \Delta g_s)|\Phi_{j\mu}^a|^2 + \Delta g_s|\Phi_{j\mu}^b|^2 \Big],
\end{aligned}
\tag{3}
$$

where $\boldsymbol{\nabla}$ is the discrete gradient, $\lambda\rho_s$ is the CDW stiffness and $g\rho_s$ is the effective CDW mass reflecting the energetic penalty for CDW ordering. The presence of such a penalty ensures that superconductivity prevails over the CDW order at $T = 0$, at least in the disorder-free regions. The mass anisotropy, encapsulated by the $\Delta g > 0$ term, is included as a result of our assumption that the potential induced by the chain layers favors ordering along the $b$ axis. Finally, and most pertinently to the present study, we assume that the application of strain causes an increase in the mass of the CDW component along the direction of the strain, while reducing the mass of the transverse component. Specifically, applying compressive strain in the $a$ direction corresponds to $\Delta g_s < 0$.

## Theoretical results

Our primary interest lies in the $k$-space (measured from $\mathbf{Q}_{a,b}$) CDW structure factor

$$
\begin{aligned}
S_\alpha(\mathbf{q}, L) = \frac{1}{N} \sum_{\mathbf{r}\mathbf{r}'} \sum_{jj'} \sum_{\mu\mu'} e^{-i\left[\mathbf{q}\cdot(\mathbf{r}-\mathbf{r}') + 2\pi\left(j - j' + \frac{\mu-\mu'}{3}\right)L\right]} \\
\times \langle \Phi_{j\mu}^\alpha(\mathbf{r})\Phi_{j'\mu'}^{*\alpha}(\mathbf{r}')\rangle,
\end{aligned}
\tag{4}
$$

where $N$ is the number of lattice points, and the averaging is over both thermal fluctuations and disorder realizations. We have used the fact that in YBCO the CuO$_2$ planes within a bilayer are separated by approximately 1/3 of the $c$ axis lattice constant.

We have calculated $S_\alpha(\mathbf{q}, L)$ by Monte Carlo simulations of Eqs. (2),(3)) on a $32 \times 32 \times 32$ (16 bilayers) system, with $\lambda = 1$, $g = 1.1$, $\Delta g = 0.1$, $\tilde{J} = 0.15$, $J = 0.015$, $\tilde{U} = 0.85$, $U = 0.12$, $V = 1$, and $\gamma = 0.15$. Each data point was averaged over 1000 disorder realizations with 8 disordered regions per bilayer and $r_d = 3$ (The disorder potential in an overlap region between two discs with different disorder orientations was taken to be the sum of the potentials. In the case of two discs with the same orientation the potential in the overlap region was randomly chosen to be one of the two.). Our simulations indicate that the qualitative trends exhibited by the model are largely insensitive to moderate changes in the model parameters. Below we indicate an instance where one can actually improve the agreement with the experimental findings by lowering the disorder strength.

Our core result is presented in Fig. 4a, which depicts the structure factor at a temperature near the superconducting $T_c$ and for various values of $\Delta g_s$, emulating the effect of $a$ axis strain. For $\Delta g_s = 0$ (absence of strain), both $S_a$ and $S_b$ show broad peaks centered around $L = 0.7$. This is a result of CDW domains that nucleate due to interaction with the disorder[39]. Since a disordered region tends to induce the same CDW pattern on both its flanking planes, an out-of-phase arrangement of the CDW order tends to form in the $c$-direction. The observed skewness away from $L = 1/2$ is a result of the form factor in Eq. (4). Increasing $\Delta g_s$ towards negative values increases the energetic cost for nucleating $a$-CDWs and leads to a decrease in the height of the corresponding peak (see inset). An opposite trend is seen for the disorder-induced peak along the $b$ direction. However, a much more dramatic effect emerges in $S_b$ beyond a characteristic value of $\Delta g_s$ in the form of a large sharp peak centered at $L = 1$. This signal originates from the regions between the disorder-induced CDW domains. It reflects the tipping of the balance between superconductivity and the $b$-CDW order in favor of the latter when the $b$-CDW nucleation cost is sufficiently reduced. The Coulomb interaction between the strain-induced CDW regions on neighboring bilayers favors an in-phase CDW configuration along the $c$-direction, hence the peak at $L = 1$. The sharpness of the peak, which corresponds to a correlation length that is of the order of the system size, is a direct result of the fact that these regions are not induced by disorder. The same is true for the in-plane correlation length, as shown in Supplementary Fig. 10.

The interplay between the superconducting and CDW orders is summarized in Fig. 4b, which shows the superconducting $T_c$ and the height of the $L = 1$ CDW peak as a function of the temperature and $\Delta g_s$. Clearly, $T_c$ (calculated by finite size analysis of the superconducting correlation length and renormalized stiffness) is a decreasing function of $|\Delta g_s|$ owing to the enhancement of the competing CDW fluctuations. A large 3D CDW signal appears for $\Delta g_s \lesssim -0.17$ at a temperature that increases with $|\Delta g_s|$, peaks in the vicinity of $T_c$ of the strained system and then rapidly diminishes at lower temperatures, as is also evident in Fig. 4c. This behavior stands in contrast to that of $S_b(0, L = 1/2)$, which similarly increases when the temperature is lowered towards $T_c$ but then saturates. Nevertheless, Supplementary Fig. 9 demonstrates that a moderate decrease in the disorder strength to $V = 0.8$ causes also

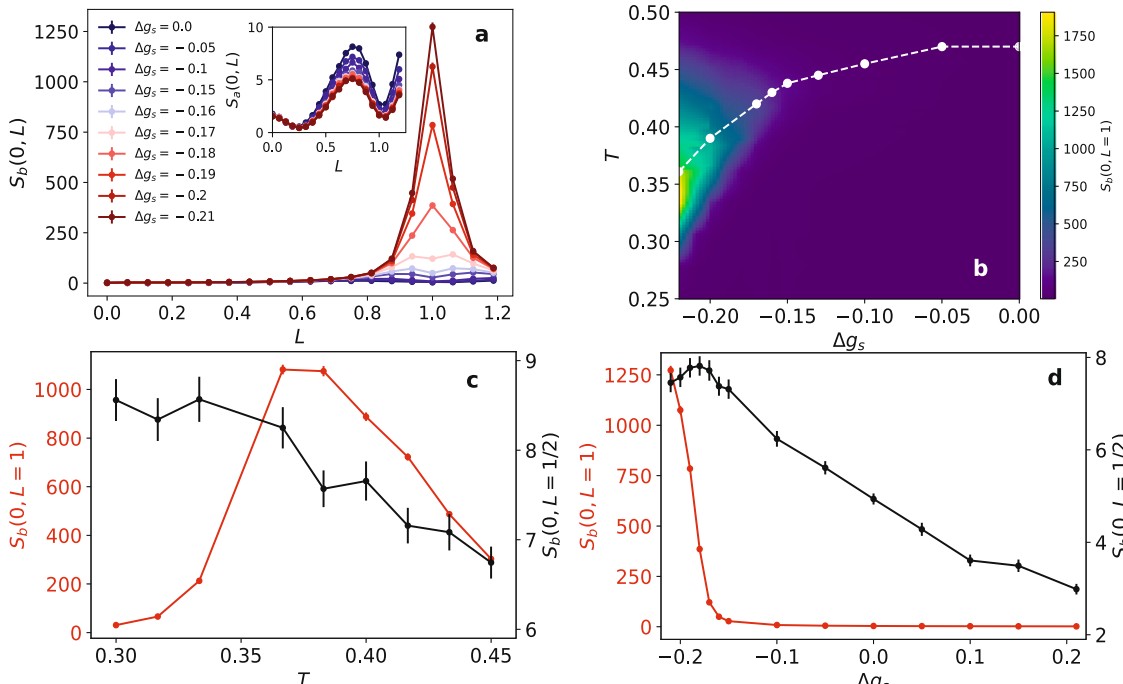

**Fig. 4 | Theoretical CDW structure factors based on a non-linear sigma model.** **a.** The CDW structure factor at the in-plane wave vector of the $b$-CDW peaks as a function of $L$ for $T = 0.383$ and various levels of the change in the CDW-mass, $\Delta g_s$, that models the effect of $a$ axis compression. The inset depicts the CDW structure factor at the position of the $a$-peak under the same conditions. **b** The intensity of the $L = 1$ $b$-CDW peak in the $T$-$\Delta g_s$ plane. The broken line marks the superconducting transition temperature. **c** The temperature dependence of the CDW structure factor at the position of the $b$-CDW peaks for $L = 1$ and $L = 1/2$ in a system with $\Delta g_s = -0.2$. **d** The $\Delta g_s$ dependence of the same quantities at $T = 0.383$. Lines are guides to the eye. The error bars represent the standard deviation of the estimated quantities calculated using rebinning.

$S_b(0, L = 1/2)$ to exhibit a peak as a function of temperature, thus recovering the qualitative experimental trend shown in Fig. 3. Supplementary Fig. 9 shows that at the same time the decreased disorder strength does not significantly alter the other aspects of the 2D and 3D CDW signals, and in particular their dependence on strain, which we turn to discuss next.

Figure 4d depicts the $\Delta g_s$ dependence of the 2D and 3D CDW signals near $T_c$. There is a clear and sharp onset of the 3D CDW peak at $\Delta g_s \simeq -0.17$, whereas $S_b(0, L = 1)$ essentially vanishes at lower values of $a$ axis strain and for all values of $b$ axis strain (positive $\Delta g_s$). The 2D signal, $S_b(0, L = 1/2)$, on the other hand, diminishes continuously with decreasing $a$ axis strain and increasing $b$ axis strain. This is because the accompanying increase in the effective mass of the $b$-CDW successively reduces the magnitude of the CDW induced by the disorder. Interestingly, and similarly to the experimental observations, we find a saturation and then a slight downturn of the 2D CDW signal that coincides with the onset of the 3D CDW order. We also note that the model predicts that the in-plane correlation length of the 2D CDW is a very weakly increasing function of the strain, while that of the 3D CDW grows rapidly beyond the onset strain, see Supplementary Fig. 10. Such behaviour is in accord with previous experimental findings[29].

## Discussion

We start our discussion from the structure of the 3D CDW. Having access to large reciprocal space maps and intense 3D CDW peaks, we can provide strong constraints for deviations from a perfectly sinusoidal charge density modulation in the long-range ordered phase. Previous studies on Bi-based cuprates[44] have indicated that in this system the CDW is commensurate with periodicity $\lambda = 4a$ over the entire doping range where it appears. More recently, high-field NMR data have been successfully analyzed in a picture where the CDW is locally commensurate with $\lambda = 3b$. In both cases it was argued that sharp phase slips (discommensurations) at the CDW domain boundary

could shift the position at which the 3D CDW peak is observed in scattering experiments to an effectively incommensurate value[40,44]. In this case, one would expect the presence of satellite peaks at $2\delta$ from $\mathbf{q}_{\text{3D-CDW}}$ ($\delta$ being the incommensurability to the commensurate 3D CDW wavevector at $k = 1/3$)[40]. At $T_c$, for YBCO$_{6.67}$ we observe no indications for additional satellite peaks along $b^*$ coming from such sharp discommensuration (we cannot fully rule out the presence of spatially extended phase slips). We also note that the $K$-cut shown in Fig. 5b for YBCO$_{6.55}$ at $T = 45$ K $< T_c$ does not exhibit a peak at $\mathbf{q}_{\text{PDW}} = \mathbf{q}_{\text{3D-CDW}}/2$. Such a peak is expected if the observed CDW is due to a PDW, i.e., a modulated superconducting condensate, at wavevector $\mathbf{q}_{\text{PDW}}$, and if the PDW coexists with uniform superconductivity[41,45]. Based on our measurements, the PDW signal must be at least 50 times smaller than the signal of the long-range 3D CDW. However, further calculations are needed to estimate the associated periodic lattice displacement and the resulting x-ray intensity to exclude the presence of PDWs.

Next, we briefly discuss our structural results in the context of nematicity, largely associated with the growth of spin and charge-density-wave correlations upon cooling[46–50] in the cuprates. In the unstrained orthorhombic unit cell the buckling angles along $a$ and $b$ axes differ only slightly. The increasing trend of the $a$ axis buckling angle under $a$ axis compression (although the error bars are relatively large compared to this subtle effect) is in line with the anisotropic modulation of the copper-oxygen-bonds with significant out-of-plane displacements of the oxygen transverse to the 2D CDW modulation inferred from zero field x-ray diffraction[51], and can be regarded as a structural parameter possibly related to nematicity. The fact that this anisotropy appears maximal in the 3D CDW phase, is also consistent with a larger response of the relative quadrupolar splitting in $^{17}$O-NMR than in $^{63}$Cu-NMR when the 3D CDW is induced using high magnetic fields[40]. The buckling anisotropy might therefore be seen as a necessary (but not sufficient) condition for 3D ordering in YBCO, but since the 3D CDW breaks both rotational and translational symmetry

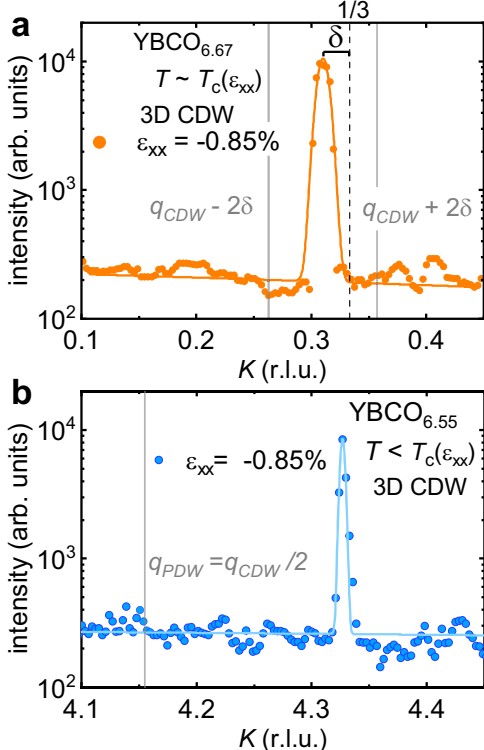

**Fig. 5 | $K$-cuts of the strain-induced 3D CDW. a** YBCO$_{6.67}$$K$-cut through the 3D CDW peak at integer $L$, maximal compression and $T = 60.5$ K - $T_c$ show an intense peak at incommensurate $K_{CDW} = 0.310(1)$ r.l.u., with $\delta$ marking the incommensurability with respect to the closest commensurate value, $K = 1/3$. Vertical grey lines mark where discommensurations would lead to satellite peaks at $K_{CDW} \pm 2\delta$. **b** YBCO$_{6.55}$$K$-cut through the 3D CDW peak at integer $L$, maximal compression and $T = 45$ K < $T_c$ in the superconducting state, where the vertical grey line marks where pair--density waves are expected to produce a peak at $K_{PDW} = K_{CDW}/2 = 0.327(1)/2$ r.l.u. Lines are Gaussian fits on top of a linear background. Note that the width of the peaks in these figures are limited by the experimental resolution (see Supplementary Information).

globally, it should not be considered as nematic[52]. In any event, to better understand the constraints imposed by the crystal structures on the formation of long-range charge order in the cuprates (in particular given the recent reports on the overdoped side[53,54]), more systematic structural investigations are needed.

We now turn to the nature of the interplay between the CDW orders and superconductivity. On the one hand, a large body of experiments has unambiguously established that the 2D CDW and superconducting orders compete in YBCO[9,10,42], as both the CDW peak amplitude and the CDW domain correlation length decrease in the superconducting state. On the other hand, in x-ray scattering, the strongest 3D CDW signals are observed where the homogeneous superconducting phase has been suppressed either by application of a large magnetic field[32–35] or by uniaxial pressure[29,30]. In both cases, the maximal achievable strength of the applied perturbations was not sufficient to completely suppress the superconducting state at the lowest temperatures. However, while in the case of strain tuning the low-temperature superconducting phase is homogeneous with no trace of the 3D CDW, the mixed state of the type-II superconductor in a magnetic field contains halos of 3D CDW around vortex cores.

In the experimental data presented above, we have shown that a form of competition can also manifest itself between the 2D and 3D CDW orders, which is best evidenced in the region of the strain-temperature space where superconductivity has been suppressed by strain. Data in unstressed YBCO$_{6.67}$ under applied magnetic field[32–35]

can also be interpreted in terms of this 2D-3D CDW competition. Specifically, it was shown in ref. 35 that as YBCO$_{6.67}$ is cooled under strong applied field there is a temperature range where the 2D CDW intensity starts to decrease while the 3D CDW keeps increasing. In the same study, a direct conversion of the 2D CDW to the 3D CDW was hypothesized, but the data there do not allow clean disentanglement of the mutual interaction among the 2D CDW, 3D CDW, and superconductivity. Our data clearly indicate that the 2D CDW stops growing and even shrinks at the onset of the 3D CDW, where the superconductivity would have set in, in the absence of strain. In the superconducting state, the evolution of 2D CDWs is complicated by their mutual competition with the 3D CDW and the simultaneous competition with superconductivity.

Finally, we note that the different temperature- and strain-dependence of the 2D and 3D CDWs can naturally be explained in the framework of the phenomenological model developed above and is rooted in the different mechanisms that are responsible for their respective establishment. The 2D CDW nucleates in domains that are dominated by their coupling to the disorder potential that arises from the doped oxygens. Consequently, these domains continue to host the CDW even when superconductivity sets in within the intervening disorder-free regions. In contrast, the 3D CDW forms in the same intervening regions at large enough strain. By assumption, the strain reduces the energetic cost of the CDW fluctuations, thus allowing them to appear at temperatures that can even exceed the $T_c$ of the unstrained system. In turn, the growth of the 3D CDW correlations contributes, via their competition with superconductivity, to the suppression of $T_c$. Below $T_c$ the 3D CDW eventually gives way to the superconducting order, which constitutes the ground state of the clean model. If the coupling between the dopants and the CDW does not change appreciably between YBCO$_{6.55}$ and YBCO$_{6.67}$, then this picture may explain why the 2D CDW is stronger in the latter (see Supplementary Table 2), as it offers more CDW nucleation centers. Nevertheless, the precise interplay between the CDW and the doped oxygens is not fully understood[55]. Within the same picture, the coincident saturation (or shrinkage) of the 2D CDW with the establishment of strong 3D CDW can be traced to phase mismatch between the two types of CDWs. The phase of the 2D CDW within the disordered domains conforms to the local disorder arrangement, whereas the 3D CDW seeks to establish a uniform phase throughout the system. The phase mismatch along the boundaries of the disordered domains gives rise to an effective surface tension, via the CDW elastic term in Eq. (3), that consequently arrests the growth of the 2D CDW domains.

On a broader perspective, we believe that our results highlight the inherent strengths of strain tuning as a powerful approach to investigate and manipulate the intricate interplay between competing orders in quantum materials, hence presenting promising prospects for advancing our understanding of these systems.

## Methods
### Samples
High-quality single crystals of YBa$_2$Cu$_3$O$_y$ with oxygen concentrations $6.55 \leq y \leq 6.80$ were grown with a flux method and then annealed and detwinned. Their superconducting $T_c$ was measured with SQUID magnetometry[42]. The crystals were then cut and polished to needles with dimensions of $\approx 200\,\mu m \times 100\,\mu m \times 2$ mm, and were pressurized along their lengths using a Razorbill CS200T piezoelectric-driven uniaxial stress cell mounted in a helium flow cryostat. Information about the samples is summarized in Supplementary Table 1. To enable fast sample exchange, the mounting into the stress cell was done via a flexible titanium support, as displayed in Fig. 1; needles were mounted into these supports in advance, a slow, delicate process, then the supports were mounted into the cell during the beamtime. To achieve higher stresses, the selected samples were further thinned down laterally using a Xenon plasma focused ion beam (PFIB), as described in

Supplementary Note 1. We checked that this procedure did not alter their superconducting properties. The samples were originally cooled under nominal zero-strain conditions – taking into consideration also the thermal response of the device used for the measurements – and then strained at low temperatures.

## X-ray diffraction

Hard x-ray diffraction was performed at the ID15B beamline of the European Synchrotron Radiation Facility with a fixed photon energy of 30 keV ($\lambda = 0.413$ Å), that enabled us to work in transmission geometry. The scattered photons were collected using a state-of-the-art large area Dectris Eiger2X CdTe 9M hybrid photon-counting detector. Additional experimental details are given in the Supplementary Note 1 and the original data are available under[56–58]. Rigaku's CrysalisPro software[59] has been used for data reduction and to obtain reciprocal space maps. Structural refinements of the average unit cell (neglecting oxygen ordering in the chain layer) have been performed using Jana2020[60]. In spite of the large dynamical range of the detector, the very high intensity difference between the CDW features and the main Bragg reflections presents a measurement challenge. To map both features, photon flux was adjusted by varying the undulator gap[61]: high photon flux was used to study CDW superstructure peaks, while low photon flux was used to record the lattice Bragg peaks without saturating the detector. Additional details about the structure refinements are given in the Supplementary Note 3.

## Strain determination

Strains reported in this paper are derived from the Bragg peak positions, eliminating error from uncertainty in strain transmission to the sample. The determination of lattice parameters was based on integration of about 400 Bragg reflections, a number that was limited by the geometrical constraints imposed by the cryostat and the stress cell, but which is nevertheless sufficient to allow accurate determination of the three lattice constants $a$, $b$, and $c$. Therefore, we also obtain the Poisson's ratios, $\nu_{ij} \equiv -\varepsilon_{ii}/\varepsilon_{jj}$, shown in Supplementary Fig. 2. Knowledge of the Poisson's ratios is useful both as a reference for other measurements, where the scattering geometry might not allow determination of the longitudinal strain[29], and potentially as a thermodynamic probe of possible changes in the electronic structure[62]. Note that throughout this paper we use the engineering definition of strain, in which negative values denote compression.

## Data availability

The data reported in this study are available at[56–58]. The data that support the findings of this study are available from the corresponding author, upon request. Source data are provided with this paper.

## Code availability

The code used to generate the Monte Carlo results is included in the Supplementary Information.

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

## Acknowledgements

We thank M. Dušek, M. Hanfland, S. Kivelson, A. P. Mackenzie, V. Petr^íček, J. Schmalian and R. Willa for helpful discussions, A. K. Jaiswal for SQUID measurements, A. Ghiami and M. Hesselschwerdt for technical support and T. Poreba and N. Maraytta for support during the diffraction experiments at the ESRF and IQMT, respectively. Self-flux growth was performed by the Scientific Facility 'Crystal Growth' at Max Planck Institute for Solid State Research, Stuttgart, Germany. This work was supported through the funding of the Deutsche Forschungsgemeinschaft (DFG, German Research Foundation), projects 422213477 (TRR 288 projects B03 and A10) and 449386310. S.M.S. acknowledges funding by the DFG - Projektnummer 441231589, M. F. acknowledges funding by the Alexander von Humboldt Foundation and the Young Investigator Preparation Grant, H. M. L. N. acknowledges support from the Alexander von Humboldt Foundation through a Research Fellowship for Postdoctoral Researchers, K. I. acknowledges the Japan Society for the Promotion of Science Overseas Research Fellowships and I. V. acknowledges the Horizon Europe MSCA fellowship 101065694. C.W.H. acknowledges support from the Engineering and Physical Sciences Research Council (U.K.) (EP/X01245X/1). We thank the European Synchrotron Radiation Facility (ESRF) for provision of synchrotron radiation facilities under proposals number HC-4226 and HC-4865.

## Author contributions

The single crystals used in this study were grown and characterized by Y.L., S.N. under the supervision of Ma. Mi. and B.K.Mi. Me. performed the single crystal XRD and structural refinement at zero strain. The crystal preparation (polishing, FIBing) for the strain cell was done by S.M.S., I.V., H.M. L.N and K.I. I.V., S.M.S., A.-A.H., T.L., M.F., G.G. carried out the XRD experiment under strain at the ESRF. I.V. and A.-A.H. performed the structural refinement and the fine analysis of the data. Y. C. and D. O. developed the theoretical model and performed the calculations. I.V., S.M.S., D.O., C.W.H.

and M.L.T. wrote the paper with input from all coauthors. M.L.T. initiated and supervised the project.

## Funding

## Competing interests

C.W.H. has 31% ownership of Razorbill Instruments, a company that markets uniaxial pressure cells such as the one used for this study. All other authors declare no competing interests.
