## [Peer Review File · Nature Communications]

Using strain to uncover the interplay between two- and three-dimensional charge density waves in high-temperature superconducting YbCoReviewers' Comments:

Reviewer #1:

Remarks to the Author:

The manuscript by Vinograd et al. presents a detailed study of the effects of uniaxial pressure on charge-density waves (CDWs) on YBCO. The research investigates how applying strain affects the competition between the short-range two-dimensional (2D) and long-range three-dimensional (3D) CDWs in this material. Utilizing x-ray diffraction, the study explores the strain and doping evolutions of these CDWs, providing insights into their interplay with superconductivity. Key findings include how strain enhances 2D CDWs and induces 3D CDWs, and how this interaction is modeled using a nonlinear sigma model of competing superconducting and CDW orders.

The competition between 2D and 3D CDW reported by the authors sheds new light on the spatial distribution of these orders, knowing now that 3D CDW lives between disorder-induced 2D CDW domains - exactly where superconductivity develops. The toy model used by the authors to describe their observations contributes to the establishment of this important new result.

After digesting the paper, I think this study really does provide new insights into the physics of superconducting cuprates, and in particular valuable information on the origin of 2D and 3D CDWs, a nearly decade-old problem. I am therefore tempted to recommend publication in Nature Communication, however, only after the authors have considerably improved the text and narrative of the manuscript.

Indeed, I found the document tedious to read and certainly not of the quality of Nat. Comm. Some ideas are repeated too many times in the paper, and the narrative doesn't get straight to the point enough. What's more, the abstract isn't really representative of the article's main findings, which is a shame for an abstract. Personally, I think the title suffers from the same problem.

On the minor side, one might add that the model and theoretical results should be simplified in the main text for the broad readership of Nat. Comm. and the details deferred to the methods. The model and the paper as a whole would have benefited from a comparison, even side by side, of the theoretical results with the experiment in Figure 5 (whose quality also differs too much from that of the other figures in the paper).

Reviewer #2:

Remarks to the Author:

The interplay of superconductivity (SC), spin, and charge order in the cuprates (and other correlated fermion materials) is one of the forefront fields of condensed matter physics. Understanding this problem is greatly advanced by the ability systematically to tune between the different phases, rather than being restricted to accessing isolated points in the phase diagram. This paper makes an important advance in the field through (uniaxial) strain tuning. It uncovers the competition between two distinct types of charge density wave (CDW) order (2D and 3D) and SC. In addition to forefront experimental work (x-ray diffraction), there is a significant complementary theoretical component. I support its publication on Nature Communication.

The authors should clarify a few points.

[] The meaning of "y" as the oxygen stoichiometry is clear. Does the manuscript define "p", eg in "y= 6.67 (p=0.125)"?

[] Perhaps the notation ϵ_{xx} and ϵ_{yy} is standard, but would it be more clear directly to label "xx" as "a-axis" and "yy" as "b-axis"? This equivalence is made in Figure 2b.

[] The x-ray intensities are in "arb. units". Yet the axes vary from $o(10^2)$ in Figure 2ab for 3D CDW to $o(1)$ in Figure 2b for 2D CDW to $o(10^3)$ in Figure 3ab for 3D CDW. Are these different relative orders of magnitude meaningful, or arbitrary?

[] In my hard-copy print-out, the vertical dashed line and left/right designations of ϵ_{xx} and ϵ_{yy} in Figure 2b was very faint.

[] The momentum peaks are identified, eg in the Figure 3 caption, to four decimal places, $q_{cdw} = 0.3102(1)$ ". It seems the data points are considerably more granular. The accuracy is warranted by the fitting procedure?

[] The authors state "It is worth noting that this strain onset is considerably lower than that estimated in previous..." Although they provide a confirming comparison, it could be worthwhile to address possible causes of the discrepancy more fully.

[] It could be useful to provide (rough) numerical values for the correlation length (top, column 2, page 4, in comparing $y=6.67$ and $y=6.55$).

[] In discussing the theoretical model, I suggest emphasizing "r" is a two dimensional (in-plane) vector (then j labels the third dimension). This is generally clear in the subsequent discussion, but an explicit statement would be useful.

[] In Eq. (2) the superfluid stiffness appears as a parameter in the Hamiltonian. Subsequently, (col 2, page 6) the stiffness is referred to as a measured quantity whose finite size scaling analysis is used to determine T_c . Can this be clarified?

[] What is the size of the CDW order parameter? Looking at Equation (4), the structure factor S is a double sum over the pair of 2D vectors r, r' and the labels j, j' of the planes. Then S is also normalized by $1/N$, the number of lattice points. With an $N=32 \times 32 \times 32$ lattice, is the maximal conceivable value of S then N^2 (from the double) sum, divided by N, so $S = o(N) = 27000$? In Figure 5d, S shoots up to $o(1300)$ for large negative Δg_s . Would it be correct to infer then that the order parameter is roughly $\langle \Phi \rangle \sim \sqrt{1300/27000} \sim \sqrt{1/20} \sim 1/4$? I realize this is probably discussed in the cited theory papers, but some sort of text along these lines could help the reader decide if S is "big", ie the sharp rise in Figure 5d indicates long range order across the whole lattice. (Perhaps a subtle issue with the presence of disorder).

[] It would be useful to have some discussion (or pointer to a place where there is a discussion) of the detailed choices of the parameters λ , g , Δg , \tilde{J} , ..., the density of random disks, etc to assure the reader the connection to the experimental work is robust. (How much do the qualitative results depend on the precise choices?)

[] In the conclusions the authors mention that $\gamma=6.67$ is more disordered than $\gamma=6.55$. Would this be modeled by a larger V in the Hamiltonian, or a greater disk density?

Minor typographical issues:

[] Page 2, column 1:
"On the other hand it the in-plane wavelengths..."
-
"On the other hand if the in-plane wavelengths..."

[] Page 2, column 2:
"Hard x-ray diffraction was performed as function of temperature ..."
-
"Hard x-ray diffraction was performed as a function of temperature ..."

[] Page 8, column 1:
"In turn the growth of the 3D CDW correlations contribute, ..."
-
"In turn the growth of the 3D CDW correlations contributes, ..."

Reviewer #3:

Remarks to the Author:

Vinograd et al. present a detailed set of measurements on the relationship between the "3D" and "2D" charge density wave orders under the influence of uniaxial strain in the canonical cuprate YBCO. The significant new results they report is that the "2D" CDW diminishes coincident with the onset of "3D" CDW order – a feature that was not evident in earlier studies.

They also present theoretical calculations of a non-linear sigma model. The results of this model are used to argue that the "2D" CDW forms due to disorder, specifically pinning around defects, whereas the "3D" CDW that has a longer correlation length occurs inbetween "2D" CDW domains and represents a more intrinsic form of the CDW order that is stabilized by the uniaxial strain. This theory follows from a similar previously published calculation by some of the co-authors.

The measurements presented are very challenging, requiring advanced sample preparation and handling that few in the field have mastered as effectively as the authors of this paper. Moreover, the presentation of the data is clear and comprehensive, showing several samples, sample geometries and important complementary measurements on the structural distortions induced by uniaxial strain.

Given the excellence of the measurements and the substantive calculations, I support publication of the paper. While it is valuable to have confirmation of a competition between the "2D" and "3D" versions of the CDW order, this is perhaps not an unexpected result, perhaps limiting the ultimate impact of the paper. However, on balance I think the quality of the measurements provide significant

contribution.

I do however, have some questions and concerns that should be addressed prior to publication.

1. The authors speculate that the 2D CDW order forms due to pinning by disorder in the chain layer. I find this proposal possible and supported by the theoretical model presented, although not directly shown. In seeming contrast, with this model, Achkar et al. (PRL 113, 107002 (2014)) showed that varying the oxygen disorder in YBCO did not lead to a change in the 2D CDW correlation length, although it did change the CDW peak intensity. This observation is seemingly at odds with the findings of the non-linear sigma model and warrants noting in the present text. Related to this, does the correlation length of the 2D or 3D CDW vary with uniaxial strain?

2. Are the measurements strain cooled or zero strain cooled? This should be clarified in the text. Presuming the 2D CDW is stabilized by disorder pinning, but the 3D CDW is not, one might expect a difference between strain cooled and zero strain cooled measurements, possibly some hysteresis.

Reviewer #1 (Remarks to the Author):

The manuscript by Vinograd et al. presents a detailed study of the effects of uniaxial pressure on charge-density waves (CDWs) on YBCO. The research investigates how applying strain affects the competition between the short-range two-dimensional (2D) and long-range three-dimensional (3D) CDWs in this material. Utilizing x-ray diffraction, the study explores the strain and doping evolutions of these CDWs, providing insights into their interplay with superconductivity. Key findings include how strain enhances 2D CDWs and induces 3D CDWs, and how this interaction is modeled using a nonlinear sigma model of competing superconducting and CDW orders.

The competition between 2D and 3D CDW reported by the authors sheds new light on the spatial distribution of these orders, knowing now that 3D CDW lives between disorder-induced 2D CDW domains - exactly where superconductivity develops. The toy model used by the authors to describe their observations contributes to the establishment of this important new result.

After digesting the paper, I think this study really does provide new insights into the physics of superconducting cuprates, and in particular valuable information on the origin of 2D and 3D CDWs, a nearly decade-old problem. I am therefore tempted to recommend publication in Nature Communication, however, only after the authors have considerably improved the text and narrative of the manuscript.

Indeed, I found the document tedious to read and certainly not of the quality of Nat. Comm. Some ideas are repeated too many times in the paper, and the narrative doesn't get straight to the point enough. What's more, the abstract isn't really representative of the article's main findings, which is a shame for an abstract. Personally, I think the title suffers from the same problem.

On the minor side, one might add that the model and theoretical results should be simplified in the main text for the broad readership of Nat. Comm. and the details deferred to the methods. The model and the paper as a whole would have benefited from a comparison, even side by side, of the theoretical results with the experiment in Figure 5 (whose quality also differs too much from that of the other figures in the paper).

Our Reply: We thank the reviewer for their careful reading of the paper, and their overall positive evaluation of our results. In an effort to address their criticism, we have tried to improve the narrative of the paper, while also keeping in mind the point of view of Reviewer #3 that our 'presentation of the data is clear and comprehensive'. Specifically, we have rewritten the abstract and proposed a new title to the paper aiming at better reflecting the main finding of the work.

Regarding the theoretical model, we have tried to leave only the essential description of its ingredients in the main text and referred to previous works for more details. A reader who is less interested in these issues can in principle skip the 'Model' section and go directly to the section on 'Theoretical results'. However, we feel that it would be unsatisfactory to merely describe the results, without stating the assumptions that were made in formulating the model and the way by which it was analyzed.

Finally, we appreciate the suggestion of a side by side comparison between theory and experiment, which we had also discussed among the coauthors before submission. Our decision not to take this route stems from our wish to avoid creating a false impression that the model is built on a microscopic understanding of the underlying physics, which is capable of offering a quantitative description of the data. This is not the case. Rather, the model is phenomenological and aims at identifying the key ingredients that are required in order to qualitatively reproduce the experimental trends, and elucidate the reasons behind them. Therefore, we would like to refrain from suggesting any quantitative agreement between the model and experiments. To this end, we firmly believe that it is important to keep the presentation of the two apart.

Reviewer #2 (Remarks to the Author):

The interplay of superconductivity (SC), spin, and charge order in the cuprates (and other correlated fermion materials) is one of the forefront fields of condensed matter physics. Understanding this problem is greatly advanced by the ability systematically to tune between the different phases, rather than being restricted to accessing isolated points in the phase diagram. This paper makes an important advance in the field through (uniaxial) strain tuning. It uncovers the competition between two distinct types of charge density wave (CDW) order (2D and 3D) and SC. In addition to forefront experimental work (x-ray diffraction), there is a significant complementary theoretical component. I support its publication on Nature Communication.

The authors should clarify a few points.

□ The meaning of "y" as the oxygen stoichiometry is clear. Does the manuscript define "p", eg in "y= 6.67 (p=0.125)"?

Our reply: We now specify in the main text that p corresponds to the doping level.

□ Perhaps the notation ϵ_{xx} and ϵ_{yy} is standard, but would it be more clear directly to label "xx" as "a-axis" and "yy" as "b-axis"? This equivalence is made in Figure 2b.

Our reply: We acknowledge the reviewer's remark. However, the notation ϵ_{xx} for the longitudinal strain under a-axis compression is standard and appears in canonical condensed-matter physics textbooks such as Kittel, and Ashcroft and Mermin. Hence, and in order to maintain contact with these texts, we would like to keep the present notation.

□ The x-ray intensities are in "arb. units". Yet the axes vary from $o(10^2)$ in Figure 2ab for 3D CDW to $o(1)$ in Figure 2b for 2D CDW to $o(10^3)$ in Figure 3ab for 3D CDW. Are these different relative orders of magnitude meaningful, or arbitrary?

Our reply: As stated in the caption of Fig. 2b, the 2D and 3D intensities correspond to integrals over different line cuts (along K and L, respectively). To compare the intensity of the 2D and 3D CDWs one has to integrate the intensity in all three reciprocal lattice directions. We have done so at selected strain values, as described in the supplementary information. In Fig. 2b, this allows us to scale the 2D and 3D intensities such that they correspond to a comparison of total integrated intensities.

□ In my hard-copy print-out, the vertical dashed line and left/right designations of ϵ_{xx} and ϵ_{yy} in Figure 2b was very faint.

Our reply: A darker color was chosen for the dashes.

□ The momentum peaks are identified, eg in the Figure 3 caption, to four decimal places,

$q_{\text{cdw}} = 0.3102(1)$ ". It seems the data points are considerably more granular. The accuracy is warranted by the fitting procedure?

Our reply: These indeed corresponded to values obtained from the fitting procedure, but we agree that this accuracy can somewhat be challenged by the experimental data. We have limited these values to three digits after the comma, which better reflect the sampling of the data.

□ The authors state "It is worth noting that this strain onset is considerably lower than that estimated in previous..." Although they provide a confirming comparison, it could be worthwhile to address possible causes of the discrepancy more fully.

Our reply: It is important to note that the determination of strain is in general a challenging issue. This is arguably best achieved when one has direct access to the lattice parameters through a set of Bragg reflections. From the nature of the experimental set-up used (IXS or RIXS spectrometers), the Bragg reflections which were accessible in previous x-ray scattering experiments (main text references H.-H. Kim et al. Science 362, 1040 (2018) and H.-H. Kim et al. Phys. Rev. Lett. 126, 037002 (2021)) were very limited compared to the ones accessible in the present study. This restriction was either due to the scattering geometry imposed by the combination of the strain cell/cryostat assembly and the beamline design, or simply due to the incident x-ray energy (soft x-rays). To estimate the a axis strain from the Bragg peaks along the $[0,0,L]$ direction, previous studies relied on the Poisson ratio ν_{zx} . This quantity has been determined with greater accuracy and is presented in supplementary figure 2. The access to a much larger number of Bragg reflections (ca. 400), including also those along the compression direction, allows us to determine in-situ all the lattice parameters and therefore the strain with a much higher accuracy than in any of the previous reports. We have clarified this at the end of our introduction (*'It is worth noting that this onset strain (and more generally any strain reported in this work) is considerably lower than that estimated in previous x-ray scattering studies [29, 30], in which the lattice parameter along the direction of compression was not accessible and in which strain could therefore not be determined with sufficient accuracy.'*)

□ It could be useful to provide (rough) numerical values for the correlation length (top, column 2, page 4, in comparing $y=6.67$ and $y=6.55$).

Our reply: We provided these estimates in Supplementary Table II and in supplementary note 2. Note that we have suppressed the sentence that the reviewer is referring to, because we have realized that the line cuts shown in (what is now) Fig. 5 are in fact limited by the experimental momentum resolution. This resolution was different for the measurement of the $y=6.67$ and that of $y=6.55$ samples, and therefore discussing the effect of disorder on the basis of this figure is not possible and would require additional studies.

□ In discussing the theoretical model, I suggest emphasizing " r " is a two dimensional (in-plane) vector (then j labels the third dimension). This is generally clear in the

subsequent discussion, but an explicit statement would be useful.

Our reply: We have added an explicit statement that r stands for the in-plane position vector in the paragraph preceding Eq.1.

□ In Eq. (2) the superfluid stiffness appears as a parameter in the Hamiltonian. Subsequently, (col 2, page 6) the stiffness is referred to as a measured quantity whose finite size scaling analysis is used to determine T_c . Can this be clarified?

Our reply: The superfluid stiffness that appears in the Hamiltonian, Eq. (2), is the bare stiffness, which determines the superfluid response of the system at the length scale of the lattice constant. We use its value as the basic energy scale. The critical temperature, on the other hand, is determined by the renormalized stiffness that describes the response of the system to a phase twist over long wavelengths (of the order of the size of the system). We have made this distinction explicit by adding “bare” and renormalized” to “stiffness” in the paragraph following Eq. (2) and on page 6, respectively.

□ What is the size of the CDW order parameter? Looking at Equation (4), the structure factor S is a double sum over the pair of 2D vectors r, r' and the labels j, j' of the planes. Then S is also normalized by $1/N$, the number of lattice points.

With an $N=32 \times 32 \times 32$ lattice, is the maximal conceivable value of S then N^2 (from the double) sum, divided by N , so $S = o(N) = 27000$? In Figure 5d, S shoots up to $o(1300)$ for large negative Δ_g . Would it be correct to infer then that the order parameter is roughly $\langle \Phi \rangle \sim \sqrt{1300/27000} \sim \sqrt{1/20} \sim 1/4$?

I realize this is probably discussed in the cited theory papers, but some sort of text along these lines could help the reader decide if S is “big”, ie the sharp rise in Figure 5d indicates long range order across the whole lattice. (Perhaps a subtle issue with the presence of disorder).

Our reply: The estimate that is offered by the reviewer is indeed correct in the presence of long-range order, which is the case for the 3D CDW in our simulations. To better reflect the development of long-range order we have chosen to refer to the correlation length. The c -axis correlation length can be deduced from Fig. 4a and is of the order of the system size. We have added a statement to this effect at the end of the third paragraph of the “Theoretical results” section. We have also included the Δ_g dependence of the in-plane correlation length of the 2D and 3D CDWs in a new Supplementary figure 10. The data clearly shows that the 3D CDW becomes essentially ordered across the entire 32^3 lattice. Finally, the reviewer has made an astute comment regarding the effects of disorder on the establishment of long-range CDW order. Indeed, owing to the Imry-Ma criterion there can be no long-range order in less than 4 dimensions when the disorder couples directly to the order parameter. However, as discussed in PRL 119, 107002 (2017) by two of us, the oxygen disorder on the chain layers couples only to the *gradient* of the $L=1$ CDW along the c direction. This fact allows the establishment of a long-range $L=1$ CDW order at $d>2$, and in particular in our three-dimensional system.

□ It would be useful to have some discussion (or pointer to a place where there is a discussion) of the detailed choices of the parameters λ , g , Δg , J , ..., the density of random disks, etc to assure the reader the connection to the experimental work is robust. (How much do the qualitative results depend on the precise choices?)

Our reply: Our experience from studying the model, both in the present work and in previous publications, is that its qualitative behavior is insensitive to moderate changes in the parameters that go into its definition. We have added such a statement to the paragraph following Eq. (4), where we specify values of the model parameters. In fact, we have not made an attempt to fully optimize the agreement between the experimental data and the behavior of the model. For example, we show in Supplementary figure 9 that by reducing the disorder potential one recovers a peak in the temperature dependence of the 2D CDW in accord with the experimental findings, while not affecting the other characteristics of either the 2D or 3D CDWs. We have also checked that the discs model, which we have used in the current study, leads to the same dependence of the CDW signals on the temperature and magnetic field, as we have found using the Gaussian noise models in PRB 92, 224504 (2015) and PRL 119, 107002 (2017). We have alerted the reader to these points at the end of the penultimate paragraph of the “Theoretical results” section and in Supplementary note 4.

□ In the conclusions the authors mention that $y=6.67$ is more disordered than $y=6.55$. Would this be modeled by a larger V in the Hamiltonian, or a greater disk density?

Our reply: We do not have a microscopic understanding of the coupling between the CDW order and the doped oxygens. As we state in our reply to Reviewer #3, and now also in the main text, the coupling can be due to local variations in the hole doping and the Coulomb interactions between the CDW and the dopants. However, if one assumes that the local environment created by the dopants is similar in the two samples then one is led to use the same V but a higher density of discs for $y=6.67$. We have added a statement to this effect to the Discussion.

Minor typographical issues:

□ Page 2, column 1:

"On the other hand it the in-plane wavelengths..." - "On the other hand if the in-plane wavelengths..."

□ Page 2, column 2:

"Hard x-ray diffraction was performed as function of temperature ..."-"Hard x-ray diffraction was performed as a function of temperature ..."

□ Page 8, column 1:

"In turn the growth of the 3D CDW correlations contribute, ..."-"In turn the growth of the 3D CDW correlations contributes, ..."

Our reply: We thank the reviewer for pointing out the typographical errors. We have corrected them as suggested.

Reviewer #3 (Remarks to the Author):

Vinograd et al. present a detailed set of measurements on the relationship between the “3D” and “2D” charge density wave orders under the influence of uniaxial strain in the canonical cuprate YBCO. The significant new results they report is that the “2D” CDW diminishes coincident with the onset of “3D” CDW order – a feature that was not evident in earlier studies.

They also present theoretical calculations of a non-linear sigma model. The results of this model are used to argue that the “2D” CDW forms due to disorder, specifically pinning around defects, whereas the “3D” CDW that has a longer correlation length occurs in between “2D” CDW domains and represents a more intrinsic form of the CDW order that is stabilized by the uniaxial strain. This theory follows from a similar previously published calculation by some of the co-authors.

The measurements presented are very challenging, requiring advanced sample preparation and handling that few in the field have mastered as effectively as the authors of this paper. Moreover, the presentation of the data is clear and comprehensive, showing several samples, sample geometries and important complementary measurements on the structural distortions induced by uniaxial strain.

Given the excellence of the measurements and the substantive calculations, I support publication of the paper. While it is valuable to have confirmation of a competition between the “2D” and “3D” versions of the CDW order, this is perhaps not an unexpected result, perhaps limiting the ultimate impact of the paper. However, on balance I think the quality of the measurements provide significant contribution.

I do however, have some questions and concerns that should be addressed prior to publication.

1. The authors speculate that the 2D CDW order forms due to pinning by disorder in the chain layer. I find this proposal possible and supported by the theoretical model presented, although not directly shown. In seeming contrast, with this model, Achkar et al. (PRL 113, 107002 (2014)) showed that varying the oxygen disorder in YBCO did not lead to a change in the 2D CDW correlation length, although it did change the CDW peak intensity. This observation is seemingly at odds with the findings of the non-linear sigma model and warrants noting in the present text. Related to this, does the correlation length of the 2D or 3D CDW vary with uniaxial strain?

Our reply: Our basic assertion is that the 2D CDW is largely nucleated due to the coupling of the CDW order to the doped oxygens on the chain layers. This may originate from induced local changes in the concentration of holes on the CuO₂ planes, as well as from the Coulomb interaction between the CDW and the dopants. We have made this point explicit in the paragraph following Eq. (2). Nevertheless, the exact details of the coupling are unknown. In particular, it might be that the development of local ortho order of the doped oxygens may actually enhance the formation of CDW in these regions as

compared to the completely disordered case. This may explain the findings of Achkar et al., who observed an increase of the CDW signal with oxygen ordering. Concomitantly, we note that even in the “ordered” phase, the correlation length of the ortho structure in Achkar’s samples was at most 6.5 lattice constants, which is only about twice the CDW period and 2.5 times smaller than the 2D CDW correlation length. This may be the reason for the apparent insensitivity of the CDW correlation length to the increase in the oxygen order. In any case, we have added a statement that reflects the uncertainty concerning the precise interplay between the doped oxygens and the CDW to the Discussion section together with a reference to Achkar’s paper.

To better characterize the strain dependence of the CDWs, we have included in Supplementary Fig. 10 the theoretical evolution of the in-plane correlation length with Δg_s . As shown by the figure, the correlation length of the 2D CDW grows very slowly with strain. On the other hand, the correlation length of the 3D CDW increases sharply beyond the onset strain for this order. These findings conform with the experimental data, as observed by Kim et al. Science 362, 1040 (2018), and we mention them at the end of the ‘Theoretical Results’ section of the revised manuscript.

2. Are the measurements strain cooled or zero strain cooled? This should be clarified in the text. Presuming the 2D CDW is stabilized by disorder pinning, but the 3D CDW is not, one might expect a difference between strain cooled and zero strain cooled measurements, possibly some hysteresis.

Our reply: In our experiments the samples were originally cooled under nominal zero-strain conditions - taking into consideration also the thermal response of the device used for the measurements - and then strained at low temperatures. We have made this point explicit in the Methods section. During the course of the measurements various temperature-strain paths were followed, but generally stress was applied at low temperatures where no indications of hysteresis were observed.

Reviewers' Comments:

Reviewer #1:

Remarks to the Author:

I thank the authors for their answers.

I recommend publication.

Reviewer #2:

Remarks to the Author:

The authors have addressed my concerns in a complete and clear way.

Reviewer #3:

Remarks to the Author:

The authors have made several useful modifications to their manuscript that aimed to clarify my concerns in the text. While there remain open questions – particularly regarding the theoretical model, I believe these are largely beyond the scope of the current paper, which retains a focus on the measurements. I do think that future work exploring theoretical model, including whether the disk-like disorder model appropriately capture oxygen disorder in the chain layer of YBCO and how do the results vary with disorder potential, may follow as future work.

Accordingly, I find the main results and arguments suitably compelling and impactful to warrant publication in Nature Communications.